# Binuclear Heteroleptic Ru(III) Dithiocarbamate Complexes: A Step towards Tunable Antiproliferative Agents

Andrea E. Gallio [1], Leonardo Brustolin [2], Nicolò Pettenuzzo [2] and Dolores Fregona [2,*]

[1] School of Chemistry, University of Bristol, Bristol BS8 1TS, UK; andrea.gallio@bristol.ac.uk
[2] Department of Chemical Sciences (DISC), University of Padova, Via Marzolo 1, 35131 Padova, Italy; leo.brus90@gmail.com (L.B.); nicolo.pettenuzzo.np@gmail.com (N.P.)
[*] Correspondence: dolores.fregona@unipd.it

**Abstract:** Binuclear dithiocarbamate complexes of Ru(III) are promising candidates in the search for outstanding metal-based anticancer agents. While different dithiocarbamates have shown ligand-dependent cytotoxicity in homoleptic binuclear Ru(III) complexes, the properties of heteroleptic analogues with different dithiocarbamate (DTC) ligands have yet to be explored. We herein propose the introduction of heteroleptic ligands as tunable features for the development of improved ruthenium-based antiproliferative agents and report a synthetic strategy for their synthesis as well as the evaluation of the cytotoxic activity of a selection of binuclear heteroleptic Ru(III) compounds towards MDA-MB-231 and PC3 cells.

**Keywords:** dithiocarbamate derivatives; ruthenium anticancer complexes; anticancer research; antiproliferative activity

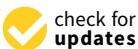



## 1. Introduction

Diverse metal-based compounds have increasingly been populating the clinical and pre-clinical arena in the chase for new chemotherapeutics matching the effectiveness of platinum-based drugs (e.g., cisplatin) but with better toxicological profiles [1–3]. In particular, a few ruthenium complexes stand out, such as NAMI-A [4], KP1019 [5], and RM- and RAPTA-Ru [6,7] which encouraged the identification of further improved ruthenium-based alternatives [8,9]. Dithiocarbamate complexes are strategically tuned towards this aim, as they combine a degree of cytoprotective action with the antiproliferative activity of metal centres [10–12]. We developed and tested several mononuclear and binuclear homoleptic dithiocarbamate ruthenium(III) complexes and identified binuclear compounds as particularly promising [13–16]. However, while ample pre-clinical studies have been conducted on binuclear homoleptic dithiocarbamate complexes of Ru(III) [13–16], it is desirable to further tune their cytotoxic activity and develop an understanding of their mechanism of action and internalization to progress their developments towards clinical trials. To achieve such an aim, we propose that a heteroleptic coordination environment may pair the cytotoxicity of well-characterized homoleptic complexes such as [Ru$_2$(pipeDTC)$_5$]Cl [13,15–17]—taken as a leading molecule in this study—with additional properties (e.g., selectivity towards specific subcellular compartments or cancerous cell lines) in complexes of the type [Ru$_2$(pipeDTC)$_x$(DTC')$_{5-x}$]$^+$ (x: 1–4; DTC' $\neq$ pipeDTC), and could be exploited for mechanistic internalization studies using fluorescent probes in the future [18].

However, no route is currently available in the literature for the synthesis of heteroleptic dithiocarbamate binuclear complexes of Ru(III).

Binuclear dithiocarbamate complexes of Ru(III) display peculiar stereochemistry and physical properties. They are typically observed as two possible isomers, $\alpha$ or $\beta$, where six sulfur donor atoms define a distorted octahedral geometry differing in the two isomers for the spatial arrangement of the ligands [13,15]. The ligand configuration in the $\alpha$

isomer describes a $\kappa^2$ or $\kappa^1$-$\mu^2$ chelating mode, while in the $\beta$ a single ligand bridges the two ruthenium centres in a $\kappa^1$-$\kappa^1$ fashion [19]. Overall, the configuration of the ruthenium atoms is, respectively, $\Delta\Lambda$ in the $\alpha$ isomer is and $\Lambda\Lambda$ in the $\beta$ isomer (Figure 1).

**Figure 1.** Reaction scheme for the synthesis of homoleptic binuclear dithiocarbamate complexes of Ru(III) via the addition of a dithiocarbamate salt to RuCl$_3$.

The Ru-Ru bond distance is short (2.743 Å, DTC = DEDT; 2.742 Å, DTC = DMDT [13,20]; [c.f. $d_{(Ru-Ru)}$ (elemental Ru; 25 °C) = 2.6502 Å, and $d_{(Ru-Ru)}$ (Ru$_3$(CO)$_{12}$) = 2.70 ÷ 2.95] [21]. The proximity between the two metals justifies the diamagnetism of these complexes by spin pairing. Assuming a perfectly $O_h$ geometry for each Ru(III) atom, which are in low-spin $d^5$ electronic configuration, a single bond involving the overlap between two T$_{2g}$ metal orbitals may account for the diamagnetism. However, diamagnetism can be also explained by considering a spin pairing effect that involves the mediation of the bridging sulfur atoms [20,22].

Below is a summary describing the traditional syntheses of binuclear homoleptic complexes.

*Synthesis of Binuclear Homoleptic Complexes of Ru(III)*

The first report on the synthesis of *tris*-(dialkyl dithiocarbamate) complexes of Ru(III) dates back to 1938 [23]. However, a method for the synthesis of *pentakis*-(dialkyl dithiocarbamate) derivatives ([Ru$_2$(DTC)$_5$]X; X: halide) was only published in the 1970s. [24] These complexes can be isolated chromatographically from a mixture of mononuclear and binuclear species obtained from the reaction of RuCl$_3$ and a dithiocarbamate salt. Interestingly, this route leads to two different binuclear isomers, namely $\alpha$ and $\beta$, that can be quantitatively converted to a pure sample of $\beta$ isomer via isomerization (Figure 1) [13,15,24].

Pignolet et al. demonstrated a method by which to selectively obtain $\beta$-[Ru$_2$(DTC)$_5$]BF$_4$ in one step by the reaction of [Ru(DTC)$_3$] with BF$_3$ under aerobic conditions [20,25]. Different *tris*-chelate metal dithiocarbamate complexes of other metal ions (M(DTC)$_3$ with M = Fe(III), Mn(II-III), Co(III) and Rh(III)) can react under similar conditions, but the main product is not easily predictable. Among these, Fe(III) results in *tris*-chelate complexes of Fe(IV) [25]. In particular, X-ray crystallography of the Fe(IV) product obtained from Fe(PyDT)$_3$ showed monomeric units with no evidence for *intra*- o *inter*-ligand interactions [26]. Similar products can also be formed when using Mn(II) or Mn(III) DTC species, leading to Mn(IV) complexes [27]. These results highlight the ability of dithiocarbamate ligands to stabilize the first transition series of the *d*-block in unusual oxidation states. On the other hand, Co(III), Rh(III) and Ru(III) generally do not lead to paramagnetic *tris*-chelate adducts of Co(IV) ($d^5$), Rh(IV) ($d^5$) and Ru(IV) ($d^4$), respectively, but products of the type [M$_2$(DTC)$_5$]BF$_4$ [20,28,29]. While cobalt and rhodium selectively lead to $\alpha$-isomers [28,29], ruthenium only forms $\beta$ isomers [20].

Finally, a route that uses [Ru(IV)(DEDT)$_3$]Cl (DEDT: diethyldithiocarbamate) as its precursor leads to $\beta$-[Ru$_2$(DEDT)$_5$]BF$_4$ by means of reaction with AgBF$_4$. Furthermore, [Ru(IV)(DEDT)$_3$]Cl is obtained by photolysis of [Ru(III)(DEDT)$_3$] in CHCl$_3$ or benzene by reaction with gaseous HCl [22,30]. However, multiple by-products are usually formed, and reproducibility can be problematic [22,30].

Starting from these methods, we set out to design a synthetic strategy for the synthesis of heteroleptic derivatives and tested their effects on cell viability and cell cycle of MDA-MB-231 and PC3 cancerous cell lines.

## 2. Results

### 2.1. A Synthetic Strategy for the Synthesis of Binuclear Heteroleptic Complexes of Ru(III)

Simple modifications of the syntheses above proved unsuccessful for the obtainment of heteroleptic derivatives. The direct addition of mixtures of different dithiocarbamate ligands to RuCl$_3$ led to a set of variously substituted binuclear heteroleptic products. In particular, heteroleptic 1:3 and beta homoleptic 2:5. Ligand exchange via the isomerization of [Ru$_2$(pipeDTC)$_5$]X (pipe: piperidine; X: Cl$^-$ or Br$^-$) as outlined in Figure 1, in the presence of differently substituted dithiocarbamates, was also ineffective. Thus, we set out to identify a suitable precursor of the type [Ru$_2$(pipeDTC)$_x$L$_y$] (x + y = 5), with leaving groups 'L' substituted by a DTC different from pipeDTC. The oxidation of the precursor [Ru(II)(pipeDTC)$_2$(NBD)] (NBD: norbornadiene) with Br$_2$ proved promising as the oxidation of Ru(II) to Ru(III) was accompanied by the substitution of NBD by the bromide anions, due to the reduced back donation capability of Ru(III) to the diene (Figure 2).

**Figure 2.** Reaction scheme for the synthesis of the bimetallic oxidized precursor.

An ESI-MS analysis showed that an oxidized precursor is a mixture of chemical species. Two bimetallic complexes were identified, namely, [Ru$_2$(III)(pipeDTC)$_3$Br$_2$]$^+$ (C$_{18}$H$_{30}$Br$_2$N$_3$Ru$_2$S$_6$$^+$) and [Ru$_2$(III)(pipeDTC)$_4$Br]$^+$ (C$_{24}$H$_{40}$BrN$_4$Ru$_2$S$_8$$^+$)—the stoichiome-

try was univocally determined via a comparison of the simulated and experimental isotopic peak patterns (Figure 3). Identical results were obtained regardless of the amount of $Br_2$ in the oxidation reaction.

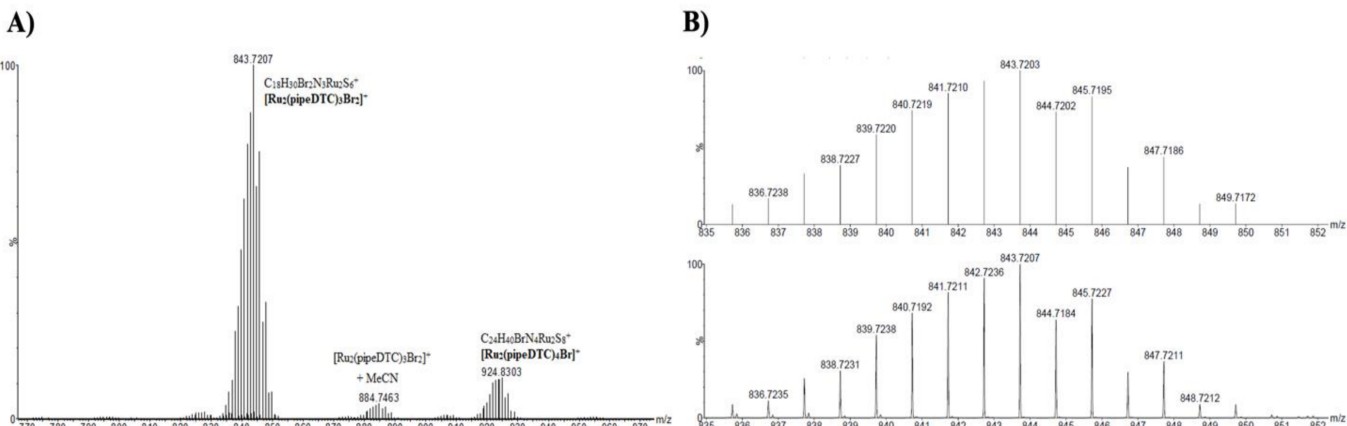

**Figure 3.** (A)ESI-MS spectra of the product obtained by oxidation of $[Ru(II)(pipeDTC)_2(NBD)]$ with $Br_2$. The parent peak at $m/z$: 843.7207 ($[Ru_2(III)(pipeDTC)_3Br_2]^+$) is accompanied by a peak ($m/z$: 884.7463) due to the adduct with the solvent (+MeCN). At $m/z$: 924.8303 a signal relative to $[Ru_2(III)(pipeDTC)_4Br]^+$ is also found. (**B**) Simulated peak pattern (above) vs. experimental peak pattern (below) for the product of oxidation of $[Ru(II)(pipeDTC)_2(NBD)]$ with $Br_2$ (oxidized bimetallic precursor $[Ru_2(III)(pipeDTC)_3Br_2]^+$). The presence of two ruthenium atoms and two bromine atoms implies a characteristic isotopic peak pattern.

Attempts to chromatographically separate the two species proved unsuccessful. However, this did not impede the further exploration and application of this synthetic strategy. We will refer to the product of the oxidation reaction of $[Ru(II)(pipeDTC)_2(NBD)]$ with $Br_2$ as the bimetallic oxidized precursor from this point on.

Interestingly, only signals within the range between 0–12 ppm were found on the $^1$H-NMR spectrum of the bimetallic oxidized precursor, highlighting a diamagnetic species resulting from an antiferromagnetic interaction between the two ruthenium atoms, possibly resulting from the bridging DTCs, bridging bromide anions and/or direct Ru-Ru bond, as confirmed by the far FT-IR spectrum.

The bimetallic oxidized precursor was used to synthesize a number of heteroleptic binuclear complexes via the direct addition of different dithiocarbamate salts. A small library of ligands was used, namely piperidine dithiocarbamate potassium salt (K(pipeDTC)), morpholine dithiocarbamate potassium salt (K(morphDTC)), indoline dithiocarbamate sodium salt (Na(indolineDTC)), *N*-methyl-*N*-propyl-carbazole dithiocarbamate sodium salt (Na(Carbz-pr-*N*(Me)-DTC)), *β*-sarcosine-naphthalene dithiocarbamate potassium salt (K(*β*-Napht-Sar-DTC)), dimethyl dithiocarbamate (DMDT), diethyldithiocarbamate (DEDT), and *β*-D-glucoside-conjugated dithiocarbamate ((gluc-MAE-DTC)). The chemical structures of the ligands are shown in Figure 4.

**Figure 4.** Chemical structures of the dithiocarbamates explored in this work: (**A**) piperidine dithiocarbamate potassium salt (K(pipeDTC)); (**B**) morpholine dithiocarbamate potassium salt (K(morphDTC)); (**C**) Indoline dithiocarbamate sodium salt (Na(indolineDTC)); (**D**) N-methyl-N-propyl-carbazole dithiocarbamate sodium salt (Na(Carbz-pr-N(Me)-DTC)); (**E**) β-sarcosine-naphtalene dithiocarbamate potassium salt (K(β-Napht-Sar-DTC)); (**F**) dimethyl dithiocarbamate (DMDT); (**G**) Diethyl dithiocarbamate (DEDT); and (**H**) β-D-glucoside-conjugated dithiocarbamate (gluc-MAE-DTC).

## 2.2. ESI-MS Characterization

An ESI-MS analysis allowed for the unequivocal identification of $[Ru_2(pipeDTC)_5]^+$ and the heteroleptic products obtained from the bimetallic oxidized precursor due to the characteristic isotopic pattern of ruthenium atoms. In most cases, a mixture of heteroleptic species was detected, as summarized in Table 1. By comparing the intensities of each signal in the product mixtures, binuclear complexes bearing three pipeDTC and two DTC′ (different from the pipeDTC derivative) consistently presented the most intense peaks. The peak intensity distribution roughly resembles the relative abundance of each species. While no truly quantitative consideration can be drawn, the ionization process during the infusion does not heavily influence the charge of the binuclear complexes, each of which were mono charged. Table 1 summarized the detected signals for all the heteroleptic binuclear ruthenium species synthesized, as provided above. Complexes are listed according to the stoichiometry of heteroleptic complexes ($[Ru_2(pipeDTC)_x(DTC′)_y]Br$). In most instances, the heteroleptic product was identified as a mixture of at least two species, where the ratio pipeDTC: DTC′ was found to be variable and appears to be ligand-dependent. However, it was possible to directly obtain $[Ru_2(pipeDTC)_3(morphDTC)_2]Br$ as a single heteroleptic species. Moreover, contrary to all other examples, the heteroleptic mixture $[Ru_2(pipeDTC)_3(gluc-MAE-DTC)_2]Br + [Ru_2(pipeDTC)_4(gluc-MAE-DTC)]Br$ could be resolved chromatographically in a later stage.

**Table 1.** ESI-MS signals for [Ru$_2$(pipeDTC)$_5$]Br and the binuclear heteroleptic Ru(III)-DTC complexes synthesized from the bimetallic oxidized precursor. Calculated (in bracket) and experimental values are listed in the right column. Typically, most of the signals are accompanied by adducts with the eluent.

| Binuclear Complex | Detected Binuclear Complex | Empirical Formula | *m/z* |
|---|---|---|---|
| [Ru$_2$(pipeDTC)$_5$]Br | [Ru$_2$(pipeDTC)$_5$]$^+$ | C$_{30}$H$_{50}$N$_5$Ru$_2$S$_{10}^+$ | (1003.9365) 1003.9354 |
| [Ru$_2$(pipeDTC)$_x$(morphDTC)$_y$]Br | [Ru$_2$(pipeDTC)$_3$(morphDTC)$_2$]$^+$ | C$_{28}$H$_{46}$N$_5$O$_2$Ru$_2$S$_{10}^+$ | (1007.8950) 1007.9023 |
| [Ru$_2$(pipeDTC)$_x$(indolineDTC)$_y$]Br | [Ru$_2$(pipeDTC)$_4$(indolineDTC)]$^+$ | C$_{33}$H$_{48}$N$_5$Ru$_2$S$_{10}^+$ | (1037.9202) 1037.9205 |
| | [Ru$_2$(pipeDTC)$_3$(indolineDTC)$_2$]$^+$ | C$_{36}$H$_{46}$N$_5$Ru$_2$S$_{10}^+$ | (1071.9054) 1071.9031 |
| | [Ru$_2$(pipeDTC)$_2$(indolineDTC)$_3$]$^+$ | C$_{39}$H$_{44}$N$_5$Ru$_2$S$_{10}^+$ | (1105.8899) 1105.8823 |
| [Ru$_2$(pipeDTC)$_x$(Carbz-pr-*N*(Me)-DTC)$_y$]Br | [Ru$_2$(pipeDTC)$_4$(Carbz-pr-*N*(Me)-DTC)]$^+$ | C$_{41}$H$_{57}$N$_6$Ru$_2$S$_{10}^+$ | (1156.9948) 1156.9969 |
| | [Ru$_2$(pipeDTC)$_3$(Carbz-pr-*N*(Me)-DTC)$_2$]$^+$ | C$_{52}$H$_{64}$N$_7$Ru$_2$S$_{10}^+$ | (1310.053) 1310.0536 |
| [Ru$_2$(pipeDTC)$_x$(β-Napht-Sar-DTC)$_y$]Br | Ru$_2$(pipeDTC)$_4$(β-Napht-Sar-DTC)]Br | C$_{38}$H$_{53}$N$_6$ORu$_2$S$_{10}^+$ | (1132.9569) 1132.9583 |
| | Ru$_2$(pipeDTC)$_3$(β-Napht-Sar-DTC)$_2$]Br | C$_{46}$H$_{56}$N$_7$O$_2$Ru$_2$S$_{10}^+$ | (1261.9784) 1261.9814 |
| [Ru$_2$(pipeDTC)$_x$(DMDT)$_y$]Br | [Ru$_2$(pipeDTC)$_2$(DMDT)$_3$]$^+$ | C$_{21}$H$_{38}$N$_5$Ru$_2$S$_{10}^+$ | (883.8422) 883.8406 |
| | [Ru$_2$(pipeDTC)$_3$(DMDT)$_2$]$^+$ | C$_{24}$H$_{42}$N$_5$Ru$_2$S$_{10}^+$ | (923.8736) 923.8732 |
| | [Ru$_2$(pipeDTC)$_4$(DMDT)]$^+$ | C$_{27}$H$_{46}$N$_5$Ru$_2$S$_{10}^+$ | (963.905) 963.9039 |
| [Ru$_2$(pipeDTC)$_x$(DEDT)$_y$]Br | [Ru$_2$(pipeDTC)$_3$(DEDT)$_2$]$^+$ | C$_{28}$H$_{50}$N$_5$Ru$_2$S$_{10}^+$ | (979.9364) 979.9362 |
| | [Ru$_2$(pipeDTC)$_4$(DEDT)]$^+$ | C$_{29}$H$_{50}$N$_5$Ru$_2$S$_{10}^+$ | (991.9365) 991.9379 |
| [Ru$_2$(pipeDTC)$_x$(gluc-MAE-DTC)$_y$]Br | [Ru$_2$(pipeDTC)$_3$(gluc-MAE-DTC)$_2$]$^+$ | C$_{38}$H$_{66}$N$_5$O$_{12}$Ru$_2$S$_{10}^+$ | (1308.0011) 1307.9977 |
| | [Ru$_2$(pipeDTC)$_4$(gluc-MAE-DTC)]$^+$ | C$_{34}$H$_{58}$N$_5$O$_6$Ru$_2$S$_{10}^+$ | (1155.9688) 1155.9681 |

### 2.3. FT-IR Spectroscopy

Furthermore, the FT-IR spectra were recorded for all Ru(III)-DTC binuclear derivatives in the 4000–600 cm$^{-1}$ and far 600–100 cm$^{-1}$ ranges. The principal band assignments are listed in Tables 2 and 3 (all FT-IR spectra are reported in Figures S15–S34). Transition metal dithiocarbamate complexes exhibit IR spectra with three main absorption regions [31–34]. The 1450–1550 cm$^{-1}$ range presents a strong band associated with the so-called "thioureide band", due to $\nu$(*N-CSS*). The presence of differently substituted dithiocarbamates and/or different coordination environments in the heteroleptic species generated multiple bands, all assigned as thioureide bands. A second absorption region accounts for $\nu$(*C-SS*) vibrations. While $\nu_a$(*C-SS*) occurs between 950–1050 cm$^{-1}$, $\nu_s$(*C-SS*) vibration is typically in the range 500–650 cm$^{-1}$. Double signals have sometimes been observed for $\nu_a$(*C-SS*) stretching, similarly to the thioureide bands. The last significant region is around 300–470 cm$^{-1}$, where $\nu$(*M-S*) can be found. All the detected signals are listed and assigned in Table 1 (950–1050 cm$^{-1}$) and 2 (500–650 cm$^{-1}$) [31–34].

**Table 2.** Collection of the main IR-vibrations (4000–800 cm$^{-1}$) of the synthesized binuclear Ru(III)-DTC complexes.

| Compounds | Vibrational Modes and Relative Frequencies (cm$^{-1}$) | | | | | | | | |
|---|---|---|---|---|---|---|---|---|---|
| | $\nu$(C=O) | $\nu_a$(C-O) | $\nu$(C=C) | $\nu$(C-H)$_{ar.}$ | $\nu$ (C-H) | $\rho$ (C-H) | $\tau$(C-H) | $\nu$ (N-CSS) | $\nu_a$ (C-SS) |
| [Ru$_2$(pipeDTC)$_5$]Br | - | - | - | - | 2928, 2851 | 850 | - | 1504, 1440 | 1000 |
| [Ru$_2$(pipeDTC)$_x$(DMDT)$_y$]Br | - | - | - | - | 2926, 2851 | 883, 851 | - | 1532, 1441 | 1000, 944 |
| [Ru$_2$(pipeDTC)$_x$(DEDT)$_y$]Br | - | - | - | - | 2971, 2928, 2851 | 849 | - | 1510, 1438 | 1000, 946 |
| [Ru$_2$(pipeDTC)$_x$(morphDTC)$_y$]Br | - | 1107 | - | - | 2933, 2853 | 878, 849 | - | 1535, 1500, 1438 | 1000, 946 |
| [Ru$_2$(pipeDTC)$_x$(indolineDTC)$_y$]Br | - | - | 1601 | - | 2930, 2851 | 851 | 748 | 1532, 1482, 1429 | 1000, 937 |
| [Ru$_2$(pipeDTC)$_x$(Carbz-pr-N(Me)-DTC)$_y$]Br | - | - | 1626, 1594 | 3047 | 2933, 2849 | 885, 849 | 750, 722 | 1503, 1483, 1441 | 1000, 998 |
| [Ru$_2$(pipeDTC)$_x$($\beta$-Napht-Sar-DTC)$_y$]Br | 1691 (NC=O) | - | 1632, 1608, 1585 | 3014 | 2935, 2851 | 886, 851, 819 | 744, 723 | 1529, 1503, 1441 | 1001 |
| [Ru$_2$(pipeDTC)$_3$(Gluc-MAE-DTC)$_2$]Br | - | 1132 (C-O-C) | - | - | 2933, 2858 | 891, 850, 798 | 723 | 1526, 1442 | 1003 |
| [Ru$_2$(pipeDTC)$_4$(Gluc-MAE-DTC)]Br | - | 1132 (C-O-C) | - | - | 2935, 2858 | 889, 858, 798 | 718 | 1524, 1507, 1443 | 1003 |

**Table 3.** Collection of the main IR-vibrations (4000–800 cm$^{-1}$) of the synthesized binuclear Ru(III)-DTC complexes.

| Compounds | Vibrational Modes and Relative Frequencies (cm$^{-1}$) | | |
|---|---|---|---|
| | $\nu_s$(C-SS) | $\nu_a$(Ru-S) | $\nu_s$(Ru-S) |
| [Ru$_2$(pipeDTC)$_5$]Br | 544 or 508 | 406 | 326 |
| [Ru$_2$(pipeDTC)$_x$(DMDT)$_y$]Br | 578, 546, 508 | 406, 365 | 325 |
| [Ru$_2$(pipeDTC)$_x$(DEDT)$_y$]Br | 573, 550, 508 | 405, 366 | 321 |
| [Ru$_2$(pipeDTC)$_x$(morphDTC)$_y$]Br | 547, 508 | 406 | 300 |
| [Ru$_2$(pipeDTC)$_x$(indolineDTC)$_y$]Br | 552, 508 | 420, 406 | 353 |
| [Ru$_2$(pipeDTC)$_x$(Carbz-pr-N(Me)-DTC)$_y$]Br | 563, 547, 508 | 437, 407 | 424 |
| [Ru$_2$(pipeDTC)$_x$($\beta$-Napht-Sar-DTC)$_y$]Br | 547, 566, 508 | 474, 406 | 366 |
| [Ru$_2$(pipeDTC)$_3$(Gluc-MAE-DTC)$_2$]Br | 549, 515, 507 | 424, 406 | 367 |
| [Ru$_2$(pipeDTC)$_4$(Gluc-MAE-DTC)]Br | n.a. | n.a. | n.a. |

Once the three signals $\nu_s$(C-SS), $\nu_a$(Ru-S) and $\nu_s$(Ru-S) relative to the homoleptic compound [Ru$_2$(pipeDTC)$_5$]Br were identified (544 cm$^{-1}$ (or 508 cm$^{-1}$), 406 cm$^{-1}$ and 326 cm$^{-1}$, respectively), the assignment of the bands due to the different DTCs was worked out by comparison.

When comparing the $\nu_a$(Ru-S) wavenumber associated with the various dithiocarbamates, an interesting trend emerges. Indeed, the stretching wavenumber decreases according to the following order: $\beta$-Napht-Sar-DTC (474 cm$^{-1}$) > Carbz-pr-$N$(Me)-DTC (437 cm$^{-1}$) > gluc-MAE-DTC (424 cm$^{-1}$) ≈ indolineDTC (420 cm$^{-1}$) > pipeDTC (406 cm$^{-1}$) = morphDTC (406 cm$^{-1}$) > DMDT (371 cm$^{-1}$) > DEDT (366 cm$^{-1}$). This series reflects the trend previously reported for the Ru-S bond strength in different dithiocarbamate complexes [16]. At the top end of the series are aromatic DTCs, where a greater degree of back-donation from the ruthenium centres to the DTC moieties can account for stronger Ru-S bonds.

Moreover, the far FT-IR spectrum of the bimetallic oxidized precursor highlights important Ru-Br vibrations. $\nu$ (*Ru-Br*) (symmetric and asymmetric stretching) in the 100–260 cm$^{-1}$ range as strong or medium intensity bands, respectively, both for bridged and terminal bromine. Bending frequencies are also found around 150 cm$^{-1}$ (not reported in Table 3; Figure S33).

*2.4. UV-Vis Spectroscopy*

The assignments for the electronic absorptions are complicated by distorted $O_h$ coordination geometries and the fact that mixtures of complexes were obtained in most cases. This led to composite bands that required deconvolution. Nonetheless, the spectra recorded for the synthesized complexes can all be divided into the following three absorption regions: (I) near the UV region, below 250 nm. The registered absorption bands are supposed to be a result of intra-ligand transitions. Two possible mechanisms can be taken into account, namely a $\pi^* \leftarrow \pi$ transition located in the N-CSS moiety and a $p \leftarrow d$ transitions between levels generated by sulfur atoms [35]. (II) Between 250–300 nm. Intra-ligand absorptions take place as well, involving $\pi^* \leftarrow \pi$ transitions of the N-C-S and S-C-S moieties, respectively. These absorptions are generally the most significant [36]. (III) Between 300–800 nm. Two bands are typically observed. The first one, of medium intensity between 330 nm and 350 nm, appears as a shoulder of the more intense intra-ligand bands. Due to its intensity, this absorption can be attributed to a metal-metal to ligand charge transfer (MMLCT) or ligand to metal-metal charge transfer (LMMCT), considering the strong interaction between the two metals [37]. The second (440–470 nm) appears to be far less intense and very broad, and is assigned as a $d \leftarrow d$ absorption [38].

All the observed bands are listed in Table 4, whereas complete spectra (800–205 nm) are reported in the SI (Figures S35–S44).

**Table 4.** Collection of UV-Vis absorption data for the synthesized binuclear ruthenium dithiocarbamate complexes in each of the three regions absorption (sh: shoulder).

| Compounds | $\lambda$ (nm) | | |
| --- | --- | --- | --- |
| | Region I | Region II | Region III |
| [Ru$_2$(pipeDTC)$_5$]Br | 209, 245 (sh) | 272 (sh), 285 | 335 (sh), 450 |
| [Ru$_2$(pipeDTC)$_3$(morphDTC)$_2$]Br | 211, 242 | 267 (sh), 289 | 339 (sh), 466 |
| [Ru$_2$(pipeDTC)$_x$(indolineDTC)$_y$]Br | 209, 244 (sh) | 266, 290 | 373 (sh), 462 |
| [Ru$_2$(pipeDTC)$_x$(Carbz-pr-*N*(Me)-DTC)$_y$]Br | 228 (sh), 235 | 260, 283, 292 | 328 (sh), 342 (sh), 445 |
| [Ru$_2$(pipeDTC)$_x$($\beta$-Napht-Sar-DTC)$_y$]Br | 211 (sh), 220 (sh), 244 | 284 | 338 (sh), 472 |
| [Ru$_2$(pipeDTC)$_x$(DMDT)$_y$]Br | 211, 223 (sh), 246 (sh) | 274 | 338 (sh), 446 |
| [Ru$_2$(pipeDTC)$_x$(DEDT)$_y$]Br | 209, 222 (sh), 248 (sh) | 267 (sh), 284 | 336 (sh), 460 |
| [Ru$_2$(pipeDTC)$_3$(gluc-MAE-DTC)$_2$]Br | 217, 242 (sh) | 287 | 338 (sh), 472 |
| [Ru$_2$(pipeDTC)$_4$(gluc-MAE-DTC)]Br | 214, 244 | 265, 286 (sh) | 339 (sh), 467 |

*2.5. $^1$H-NMR Spectroscopy*

The $^1$H-NMR spectra of [Ru$_2$(DTC)$_5$]Br are characterized by the absence of signal out of the 0–12 ppm range. All the possible ligand coordination modes account for complicated signals, which often overlapped (Figures S6–S14). It is, however, possible to identify diagnostic signals of the $\alpha$- and $\beta$-isomers, respectively, by comparing the spectra of $\beta$-[Ru$_2$(pipeDTC)$_5$]X (X: Cl$^-$, BF$_4$$^-$), obtained via the literature methods described in the introduction, with those relative to a mixture of $\alpha$ and $\beta$, (Figure 5).

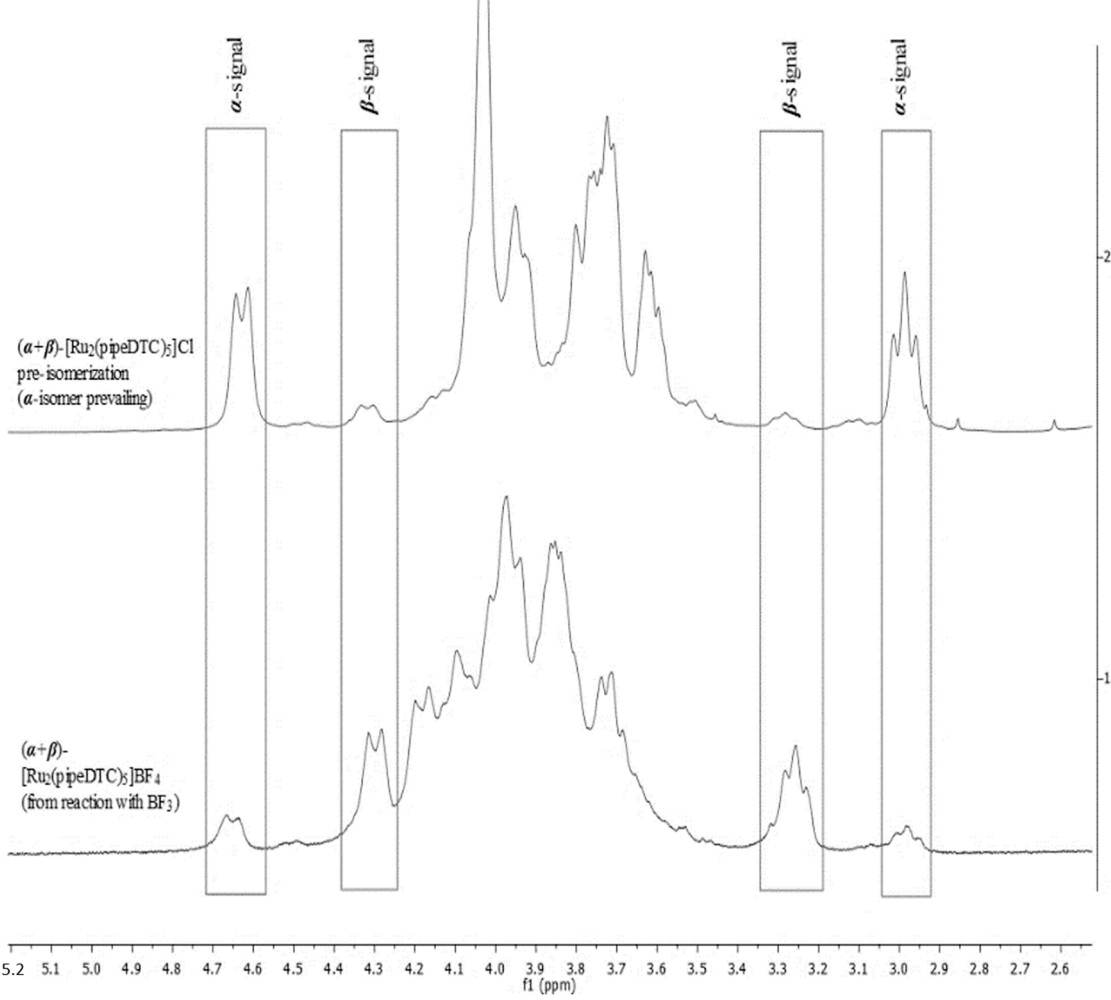

**Figure 5.** Analysis of two ¹H-NMR (600 MHz; CDCl₃) spectra of [Ru₂(pipeDTC)₅]Cl (**above**) and [Ru₂(pipeDTC)₅]BF₄ (**below**) allowed for the identification of peaks diagnostic for the α and the β isomers.

It should be noted that this approach was successfully applied only for the homoleptic [Ru₂(pipeDTC)₅]X (X: Cl⁻, Br⁻). W−ith other dithiocarbamates, the signal pattern is generally significantly more intricate. In the case of heteroleptic complexes, even more complicated spectra are observed. Especially for the reaction synthesis of binuclear derivatives starting from the bimetallic oxidized precursor, it was possible to argue that our synthetic strategy, at least in the case of [Ru₂(pipeDTC)₅]Br, favours the formation of the β-isomer (Figure 6).

To further study the preferential formation of the β-isomer, we focussed on the ¹H-NMR spectrum of [Ru₂(DMDT)₅]Br (DMDT: dimethyldithiocarbamate), which consists of five distinct singlets (one for each DMDT) and facilitates the type of comparative analysis shown in Figure 7. We synthesized [Ru₂(DMDT)₅]Br via the addition of DMDT to a bimetallic oxidized precursor with DMDT ligands obtained by oxidation of [Ru(II)(DMDT)₂(NBD)] with Br₂. Peak assignment was performed according to Wheeler and co-workers [20] and revealed that the β-isomer is indeed largely dominant (Figure 7).

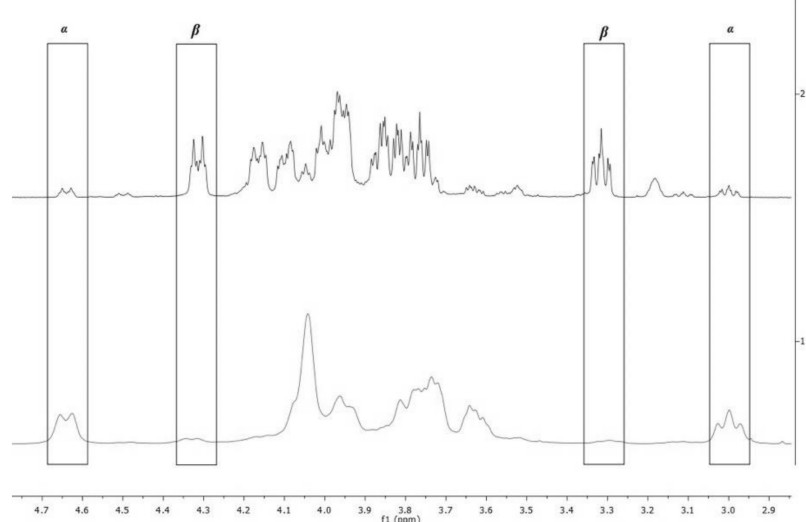

**Figure 6.** Analysis of the two $^1$H-NMR spectra of [Ru$_2$(pipeDTC)$_5$]Br obtained starting from our oxidized precursor (**above**) in comparison with the [Ru$_2$(pipeDTC)$_5$]Cl obtained by reacting RuCl$_3$(3·H$_2$O) with three equivalents of piperidine dithiocarbamate (**below**). Our new synthetic strategy shows a neat prevalence of β-[Ru$_2$(pipeDTC)$_5$]Br.

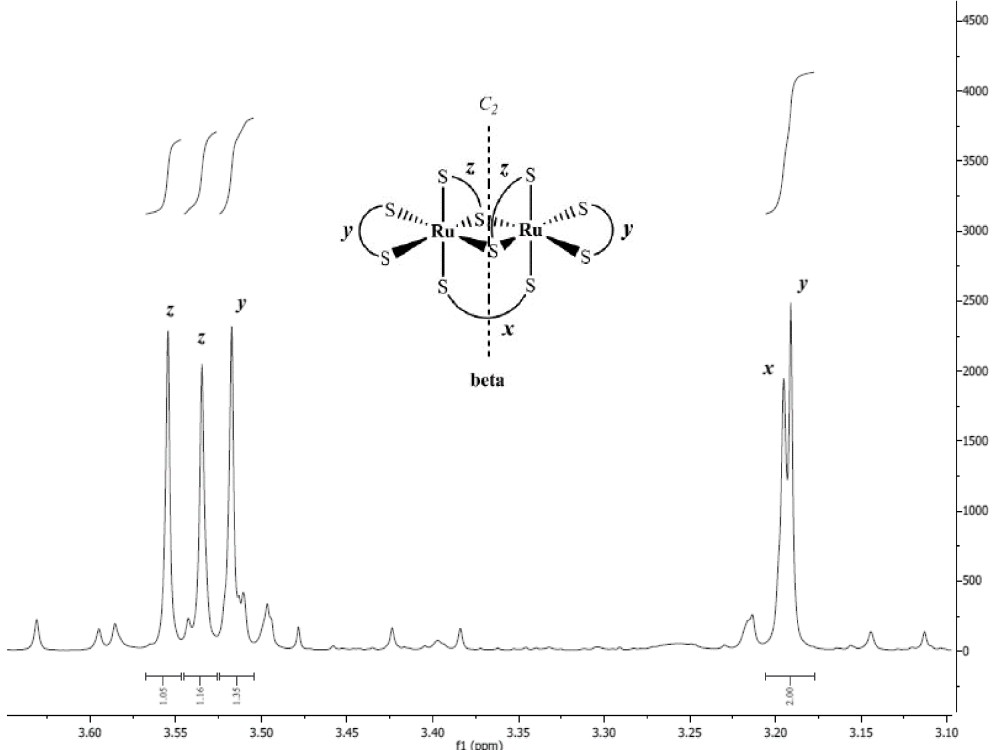

**Figure 7.** $^1$H-NMR spectrum and peaks assignment of β-[Ru$_2$(DMDT)$_5$]Br. (CDCl$_3$; 600 MHz).

The $^1$H-NMR spectra comparison between [Ru$_2$(pipeDTC)$_5$]Br, [Ru$_2$(DMDT)$_5$]Br and [Ru$_2$(pipeDTC)$_x$(DMDT)$_y$]Br reveals whether each group of signals is relative to the co-ordinated pipeDTC or DMDT, respectively. In particular, two singlets in the spectrum of [Ru$_2$(DMDT)$_5$]Br at ~3.2 ppm are clearly missing in the spectrum of [Ru$_2$(pipeDTC)$_x$(DMDT)$_y$]Br (Figure 8). For β-[Ru$_2$(DMDT)$_5$]BF$_4$, Wheeler and co-workers assigned the two singlets near 3.2 ppm, respectively, to a non-bridging chelate DMDT and the single κ$^1$-κ$^1$ bridged ligand, represented, respectively, by green and the black in Figure 9 (β-configuration) [20]. Assuming, for simplicity, a perfectly octahedral environment for each Ru atom, ligands

may be indexed according to the symmetry of the $\alpha$- and $\beta$-homoleptic isomers, both of which belong to the $C_2$ point group (Figure 9).

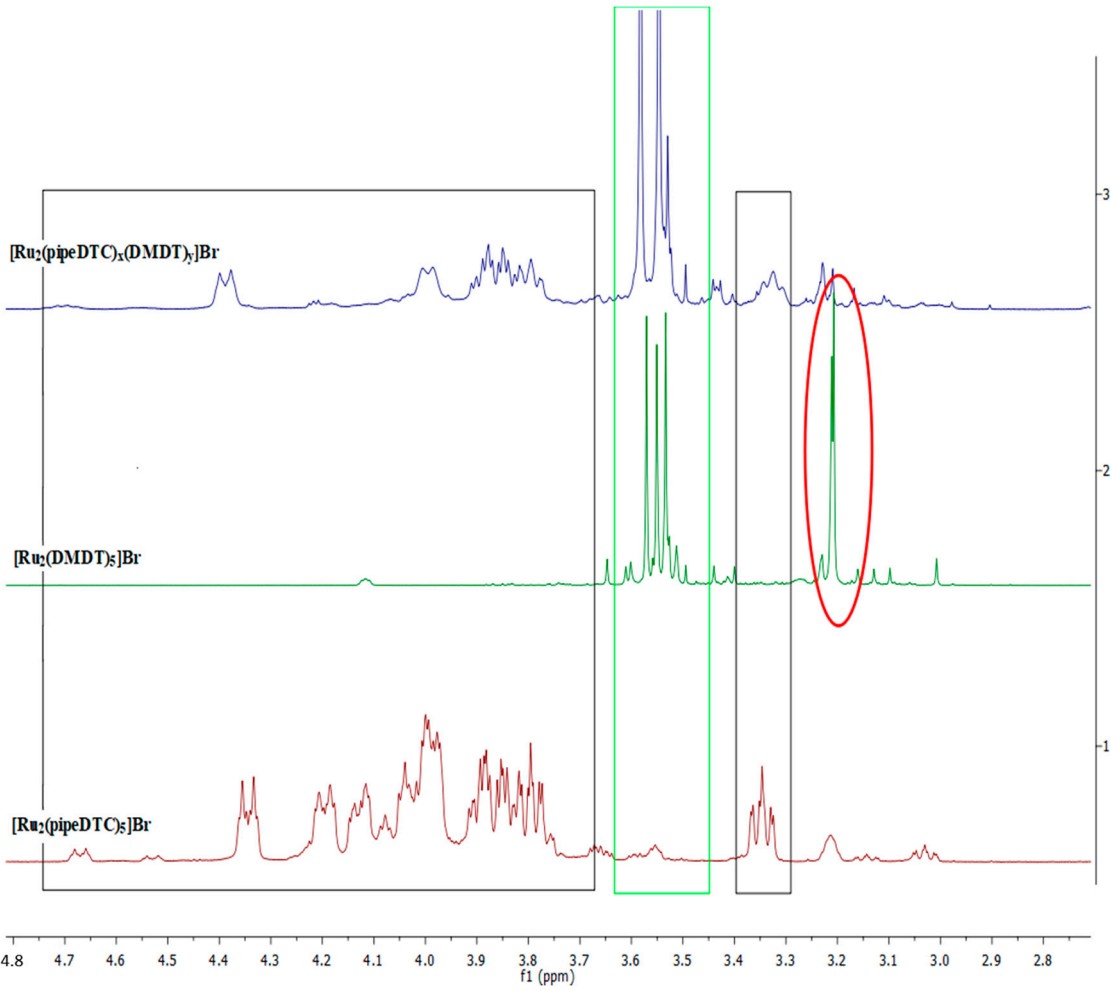

**Figure 8.** Comparison of the $^1$H-NMR spectra of $[Ru_2(pipeDTC)_5]Br$, $[Ru_2(DMDT)_5]Br$ and $[Ru_2(pipeDTC)_x(DMDT)_y]Br$. In the spectrum of $[Ru_2(pipeDTC)_x(DMDT)_y]Br$ some resonances related to coordinated DMDT are not represented (red circle). The remaining DMDT signals are within the green rectangle, whereas the black boxes emphasise the coordinated pipeDTC signals.

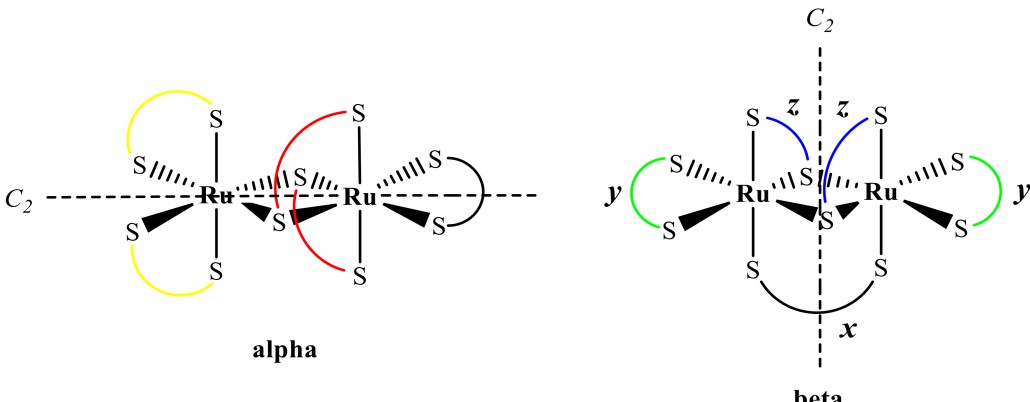

**Figure 9.** Symmetry equivalent coordinated DMDTs indexed by colour both in the $\alpha$- and $\beta$-configurations (assuming an overall $C_2$ symmetry. The colour scheme refers to the discussion in the main text.

In the absence of crystallographic data, this indicates that the "x" and "y" positions in Figure 7 are not occupied by DMDT ligands in the coordination sphere of the two ruthenium atoms in $[Ru_2(pipeDTC)_x(DMDT)_y]Br$.

Overall, the above considerations lead us to conclude that the $\kappa^1$-$\kappa^2$ coordination modes (represented in blue in Figure 9) are the most easily assumed by the entering DTC' within the unknown mechanism of reaction involved in our synthetic strategy.

Nevertheless, the discussion above will have to be followed up by quantum-mechanical calculations before reaching any rigorous conclusion.

### 2.6. Antiproliferative Activity

The homoleptic complex $[Ru_2(III)(pipeDTC)_5]Br$ (from here on compound #1) and the heteroleptic complexes $[Ru_2(pipeDTC)_3(morphDTC)_2]Br$ (compound #2) and $[Ru_2(pipeDTC)_3(gluc\text{-}MAE\text{-}DTC)_2]Br$ (compound #3) were selected to study their antiproliferative properties. Homoleptic compound #1 was taken as a benchmark to investigate the effect of heteroleptic ligands in the coordination environment of these binuclear ruthenium dithiocarbamate complexes.

Two types of adenocarcinomas were tested, namely the human triple-negative breast cancer MDA-MB-231 and the PC3 prostate androgen-resistant cancer. The MDA-MB-231 cells belong to the so-called TNBC subtype as they do not express the genes for the estrogen receptor (ER), the progesterone receptor (PR) and the human epidermal growth factor receptor 2 (HER2/neu), making traditional hormonal or monoclonal antibody therapies ineffective [39,40]. Specific to the PC3 prostate cancer cell line is androgen unresponsiveness, which rules out the benefits of androgen ablation in the early stages of the disease. Therefore, these types of androgen-independent tumours require alternative therapeutic strategies, as breast and prostate cancer are amongst the deadliest forms of tumour [41].

The MDA-MB-231 and PC3 cells were exposed to different concentrations of the metal-DTC complexes (i.e., 0.05, 0.1, 1.5 and 10 μM) for 24 and 48 h, after which cell viability was measured, and cell cycle analysis was performed.

#### 2.6.1. Cell Viability

Cell viability was measured either by the Trypan Blue or WST-1 colourimetric assays in three independent experiments performed in triplicate. Viability assays were then performed 24 and 48 h after treatment for each compound following standard experimental procedures. Three independent experiments were carried out for each compound. The IC50 values were obtained by fitting the experimental logarithmic viability curves obtained from the cell count after 24 and 48 h of treatment with the compounds under examination (Figures S46–S51 and S56–S61). The curves were calculated after normalizing the values obtained at the different concentrations with the value of the control ("untreated"). In addition, column graphs of the number of living and dead cells in the different experimental conditions were obtained (Figures S52–S55).

The IC50s are summarized in Table 5. Further experimental details are reported in the SI.

**Table 5.** IC50 values calculated either via Trypan Blue or WST-1 tests 24 and 48 h post-treatment of MDA-MB-231 or PC3 cells with compounds #1, #2, and #3, respectively.

| | $IC_{50}$ (μM) | | | | | | | |
|---|---|---|---|---|---|---|---|---|
| **Viability Test** | **Trypan Blue** | | | | **WST-1** | | | |
| **Cell Line** | **MDA-MB-231** | | **PC3** | | **MDA-MB-231** | | **PC3** | |
| **Treatment Time** | **24 h** | **48 h** | **24 h** | **48 h** | **24 h** | **48 h** | **24 h** | **48 h** |
| Compound #1 | $0.9 \pm 0.3$ | $1.0 \pm 0.1$ | $1.05 \pm 0.12$ | $0.66 \pm 0.05$ | $0.6 \pm 0.1$ | $0.6 \pm 0.1$ | $1.1 \pm 0.1$ | $0.4 \pm 0.2$ |
| Compound #2 | $1.3 \pm 0.2$ | $0.25 \pm 0.1$ | $1.17 \pm 0.13$ | $0.27 \pm 0.09$ | $0.39 \pm 0.1$ | $0.25 \pm 0.3$ | $1.47 \pm 0.3$ | $1.08 \pm 0.2$ |
| Compound #3 | N.A | ~1.6 | N.A. | N.A. | $0.01 \pm 0.02$ | $0.12 \pm 0.05$ | N.A | N.A. |

Effects of the Treatments on Cell Growth of Breast Adenocarcinoma Line MDA-MB-231

Treatment with compounds #1 and #2 induced, compared to the control, significant effects on cell growth starting at as early as 24 h. The IC50s calculated by Trypan blue assay were approximately equal to 1 µM for both experimental times for compounds #1, while those calculated by the assay with the WST-1 are slightly lower (0.66 µM). For compounds #2, the IC50 calculated for the two experimental times are around 1 µM at 24 h and 0.25 µM at 48 h (Trypan blue assay). Slightly lower values were obtained using the WST-1 test for the two experimental times. Overall, advanced necrosis was induced by compounds #1 and #2 within 24 h post-treatment. Unlike compounds #1 and #2, the Trypan blue data shows that compound #3 reduces the number of viable cells by only 60% within 24 h at the highest concentration tested (10 µM). This percentage is slightly higher at 48 h, with a calculated IC50 of about 1.6 µM. (Figure S48). However, from the WST-1 results, even the lowest tested concentration of the compound did not trigger any bio-reduction of WST-1. The tetrazolium salt (WST-1) was cleaved by mitochondrial dehydrogenases into formazan, which was soluble and intensely coloured. Such bio-reduction relies on the bioavailability of NADPH. Therefore, the amount of formazan that is formed correlates with the number of metabolically active cells in the culture. We reasoned that the treatment could have given rise to an alteration in the metabolic process that leads to a reduction in glycolytic production of NADPH without inducing immediate cell death. This hypothesis is also supported by the results in Figures S52 and S53, where only at the highest concentration is there a significant decrease in the number of viable cells, while the dead cells count remains constant.

Effects of the Treatments on Cell Growth of the PC3 Prostate Cancer Line

The treatments with compounds #1 and #2 induced significant effects on cell growth. Furthermore, the IC50s calculated for compound #1 by the Trypan blue assay at 24 and 48 h are 1.05 µM and 0.66 µM, respectively. The IC50 values for compound #2 are around 1.17 µM at 24 h and 0.27 µM at 48 h. These values are comparable to those obtained from the WST-1 test for compound #1, whereas they are slightly higher and equal to 1.47 µM and 1.08 µM for compound #2, respectively, at 24 and 48 h. Contrary to compounds #1 and #2, the effect of compound #3 on the growth of these cells was even more limited than that observed on the breast adenocarcinoma line. We can note that in PC3, there is a decrease in cell viability only at the highest concentration. This decrease never reaches 50% as reported in the column charts (Figures S54 and S55).

2.6.2. Cell Cycle Studies

Subsequently, we evaluated the effect of the three compounds on the cell cycle of the MDA-MB-231 and PC3 cell lines. Cells were exposed to compounds #1 and #2 diluted in the cell culture wells to the IC50 concentrations obtained with Trypan blue test at 24 and 48 h, respectively (Table 5). A final concentration of 20 µM was instead used for compound #3. This is justified by the small quantity of available compound following previous cytotoxicity experiments and the difficulties we encountered in precisely weighing the compound, with it being highly electrostatic. At 24 and 48 h after treatment with the compounds, the percentages of the various phases of the cell cycle were evaluated by cytofluorimetric analysis (methods in SI). Tables 6 and 7 show the mean ± SD of the percentages of the various phases of the cycle of the two cell lines, respectively. Figures in SI (Section S1) show all the histograms at various times and with various treatment conditions on both cell lines for cell cycle experiments.

**Table 6.** Averages ± SD of the measured percentages of the cell cycle in MDA-MB-231 24- and 48 h post-treatment with control (vehicle), compound #1, compound #2, and compound #3. Pre-G0/G1 figures are omitted.

| % of the Cell Cycle Stages in MDA-MB-231 | | | | | |
|---|---|---|---|---|---|
| $G_0/G_1$ | | S | | $G_2/M$ | |
| Time 0 | 44.5 ± 4.6 | | 39.8 ± 0 | | 12.05 ± 0.3 | |
| Treatment time | 24 h | 48 h | 24 h | 48 h | 24 h | 48 h |
| Control | 48.0 ± 1.5 | 59.1 ± 0.7 | 39.1 ± 0.1 | 32.2 ± 1.1 | 11.6 ± 1.0 | 13.05 ± 0.1 |
| Compound #1 | 21.9 ± 1.3 | 13.0 ± 1.0 | 43.5 ± 3.8 | 31.3 ± 1.7 | 25.0 ± 4.5 | 17.95 ± 0.9 |
| Compound #2 | 22.4 ± 1.2 | 17.7 ± 1.4 | 51.2 ± 0.3 | 32.9 ± 0.5 | 20.7 ± 0.2 | 15.15 ± 1.2 |
| Compound #3 | 47.8 ± 5.4 | 63.7 ± 3.8 | 27.7 ± 6.8 | 25.1 ± 0.2 | 19.8 ± 4.3 | 16.35 ± 0.2 |

**Table 7.** Averages ± SD of the measured percentages of the cell cycle in PC3 24- and 48 h post-treatment with control (vehicle), compound #1, compound #2, and compound #3. Pre-G0/G1 figures are omitted.

| % of the Cell Cycle Stages in PC3 | | | | | |
|---|---|---|---|---|---|
| $G_0/G_1$ | | S | | $G_2/M$ | |
| Time 0 | 39.3 ± 5.3 | | 33.4 ± 2.5 | | 18.2 ± 3.0 | |
| Treatment time | 24 h | 48 h | 24 h | 48 h | 24 h | 48 h |
| Control | 24.3 ± 12.3 | 28.1 ± 2.3 | 46.4 ± 7.5 | 43.2 ± 1.5 | 19.1 ± 3.0 | 19.8 ± 1.0 |
| Compound #1 | 40.1 ± 2.9 | 32.3 ± 0.6 | 31.4 ± 1.0 | 27.0 ± 0.1 | 16.7 ± 0.1 | 14.2 ± 0.6 |
| Compound #2 | 42.2 ± 1.3 | 32.7 ± 2.5 | 37.4 ± 4.0 | 26.4 ± 0.9 | 7.3 ± 6.2 | 10.7 ± 1.0 |
| Compound #3 | 31.9 ± 2.7 | 28.7 ± 2.2 | 37.6 ± 2.7 | 36.9 ± 1.5 | 20.0 ± 1.0 | 20.1 ± 2.2 |

Effects of Treatment on the Cell Cycle of the Breast Adenocarcinoma Line MDA-MB-231

For compounds #1 and #2 at 24 h, compared to the control, an increase in the percentage of cells in the S and G2/M phases, a decrease in the percentage of cells in the G0/G1 phase and a beginning of the accumulation of cells at the pre-G0/G1 peak were observed. The decrease in the percentage of cells in the G0/G1 phase is greater at 48 h, where the accumulation of cells at the pre-G0/G1 peak is much more prominent and indicates the presence of intensive apoptosis/necrosis.

In Figure 10, the effect of compound #1 on MDA MB-231 cells after 24 and 48 h of treatment is reported as an example. Cell cycle histograms for all the tested conditions are reported in SI (Section S1).

It can be seen how the treatment with the compound, even at 24 h, induces the appearance of a preG0/G1 peak (arrow). This peak increases considerably at 48 h. The appearance of the pre-G0/G1 peak denotes the presence of cells in the apoptotic phase (apoptotic bodies) with DNA content <2n.

Compound #3 compared to the control induced a decrease in the S phase and a slight increase in the G2/M phase at 24 h. This effect also lasts for 48 h. It must be remembered that the concentration of this compound used is about 20 times higher than that of compounds #1 and #2.

Effects of Treatment on the Cell Cycle of the PC3 Prostate Cancer Line

A 24 h treatment of the cell line with compounds #1 and #2 determined an increase in the percentage of cells in the G0/G1 phases, a decrease in the percentage of cells in the S and G2/M phase and an accumulation of cells at the pre-G0/G1 phase. This characteristic is also visible at 48 h, except for with the pre-G0/G1 peak, which is much more prominent. Compared to the control, Compound #3 induces only a slight decrease in the S phase in both experimental times.

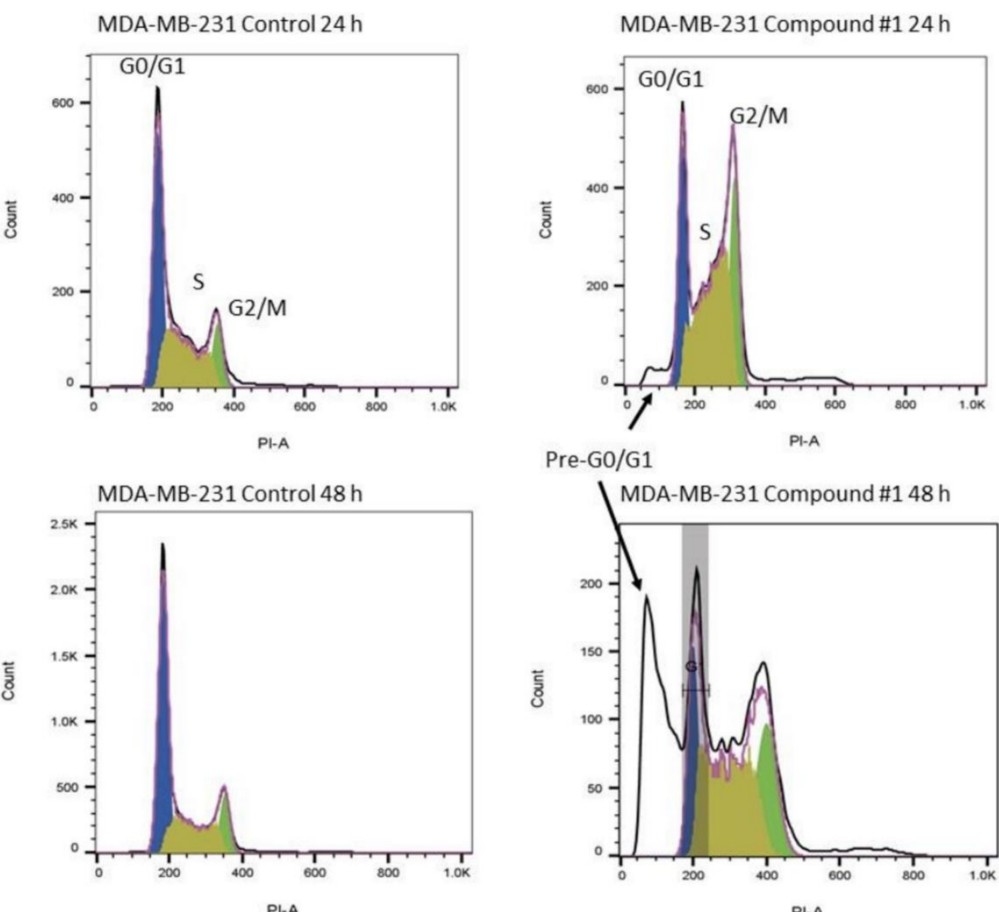

**Figure 10.** Histograms representing the cell cycle stages (pre-G0/G1, G0/G1, S, and G2/M) in MDA-MB-231 24 and 48 h in the presence of compound #1 (**right**) compared to the control (**left**).

## 3. Discussion

In the present work, we produced a library of binuclear heteroleptic ruthenium dithio-carbamate complexes. However, only the morphDTC and gluc-MAE-DTC ligands allowed for the obtainment of stoichiometrically pure products, respectively, $[Ru_2(pipeDTC)_3(morphDTC)_2]Br$, $[Ru_2(pipeDTC)_3(gluc-MAE-DTC)_2]Br$ and $[Ru_2(pipeDTC)_4(gluc-MAE-DTC)]Br$. In particular, the two gluc-MAE-DTC derivatives were chromatographically isolated from a mixture of the two, while morphDTC led selectively to the doubly substituted species. All the other synthesized heteroleptic complexes were obtained as a mixture of products containing at least two derivatives, that could not be purified chromatographically but which were unequivocally identified by ESI-MS. The ESI-MS evidence on the nature of the pipeDTC-oxidized precursor, containing both the $[Ru_2(III)(pipeDTC)_3(Br)_2]Br$ and the $[Ru_2(III)(pipeDTC)_4Br]Br$ derivatives, can justify the obtainment of the heteroleptic products of stoichiometry $[Ru_2(III)(pipeDTC)_3(DTC')_2]Br$ and $[Ru_2(III)(pipeDTC)_4(DTC')]Br$, respectively. The presence of otherwise substituted heteroleptic complexes in some products may be due to rapid substitution equilibria involving a neo-formed heteroleptic binuclear complex and the free DTC' in solution. The nature of the entering ligand appears to be crucial in determining the final distribution of products.

Trypan blue and WST-1 tests for $[Ru_2(III)(pipeDTC)_3(morphDTC)_2]Br$ (compound #2) showed high cytotoxicity towards MDA-MB-231 and PC3 cells, comparable to that of $[Ru_2(III)(pipeDTC)_5]Br$ (compound #1), implying that the introduction of two morphDTC ligands does not substantially affect the overall biological activity of the homoleptic complex. This is remarkable considering the antiproliferative inactivity of $[Ru_2(III)(morphDTC)_5]Cl$ for similar cell lines [15]. On the other hand, $[Ru_2(pipeDTC)_3(gluc-MAE-DTC)_2]Br$ (compound #3), even at the highest concentration at which it was tested, did not show as

significant a toxicity on both MDA-MB-231 and PC3 cells. This may be indicative of a quenching effect introduced by the glycosidic ligands to the overall anti-proliferative properties of compound #1. Moreover, WST-1 results for compound #3, indicates an alteration of the metabolic process in MDA-MB-231 with a consequent reduction in the glycolytic production of NADPH by the cells.

The cell cycle analysis suggests the triggering of apoptotic/necrotic processes as early as 24 h post-treatment. This phenomenon is emphasized after 48 h of incubation, especially in the breast adenocarcinoma line MDA-MB-231.

## 4. Materials and Methods

### 4.1. Materials

Analytical TLCs were performed using Kiesegel F254. UV light ($\lambda$ = 254 nm). Gravity column chromatography was performed on Silica gel 60 (0.063–0.200 mm, 70–230 mesh) (Merck). Flash liquid chromatography was performed using a semiautomatic liquid chromatography system (Biotage; SPX software 2.1; UV-vis detector set to 254 nm). In this latter case, disposable silica gel- or inverse phase-cartridges were chosen accordingly to the nature and the amount of sample to be purified. ZIP-type, SNAP-type and C18-cartridges were directly purchased from Biotage. The elution of the loaded compounds was obtained by using the proper eluent mixture. Elemental analyses were carried out at the Microanalysis Laboratory of the Department of Chemical Sciences, the University of Padova by using a microanalyzer Fisons EA-1108 CHNS-O and a microanalyzer Carlo Erba 1108 CHNS-O. Near-FTIR spectra (4000–400 $cm^{-1}$) were registered at room temperature (32 scans, resolution 2 $cm^{-1}$) by Nicolet Nexus 5SXC spectrophotometer. Samples were prepared using KBr pellets, according to standard procedures. Far-FTIR spectra (600–100 $cm^{-1}$) were registered at room temperature (200 scans, resolutions 2 $cm^{-1}$) with a Nicolet Nexus 870 spectrophotometer. For the analysis, films of samples dispersed in nujol were deposited in polyethene discs. All spectra were processed with OMNIC 5.2 (Nicolet Instrument Corporation). $^1$H-NMR spectra of DTCs were recorded at 298 K on a Bruker Avance DRX300 spectrometer equipped with a BBI [$^1$H, X] probe-head, Bruker. Typical acquisition parameters for 1D $^1$H-NMR spectra ($^1$H: 300.13 MHz) were as follows: 64 transients, spectral width 15 ppm, using delay time of 1.0–7.0 s. $^1$H-NMR spectra of Ru-DTC complexes were recorded at 298 K on a Bruker Avance DMX600 spectrometer equipped with a Triple Resonance Probe TXI [$^1$H, $^{13}$C, $^{15}$N], 5mm, three axes gradient, Bruker. Typical acquisition parameters for 1D $^1$H-NMR (1H: 599.90 MHz) were as follows: 64 transients, spectral width 15 ppm, delay time of 1.0–7.0 s. The data sets were processed with the standard Bruker processing software Topspin 1.3. Chemical shifts were referenced to solvent signals. Peak assignment and integral calculations were carried out with MestreReNova version 12.0.0 (Mestrelab Research S.L.). The absorption spectra of freshly prepared solutions of samples were acquired at 25 °C and 37 °C in the range 800-200 nm, taking into account the solvent cut-off, with an Agilent Cary 100 UV-Vis double beam spectrophotometer. Samples were dissolved in the appropriate solvents and the resulting solutions were placed in QS quartz cuvette (path length 1 cm). ESI-MS was recorded with a positive mode on a Xevo G2S QTof (Waters) instrument, setting a 2.5 kV ionization potential. Samples were dissolved in methanol or acetonitrile, using, respectively, pure methanol and pure acetonitrile as eluent. ESI-MS spectra were processed by the software MassLynx.

### 4.2. Synthesis of the Ligands

With the exceptions of DMDT and DEDT—which were commercially available—all dithiocarbamates in this work were synthesized from the corresponding amines by reaction with $CS_2$. $^1$H-NMR spectra of the synthesized ligand are reported in Figures S1–S5.

#### 4.2.1. Piperidine Dithiocarbamate Potassium Salt

Piperidine was stoichiometrically added to an EtOH solution of KOH (100 mmol). An excess of $CS_2$ (200 mmol) was added dropwise. The reaction mixture was kept under

vigorous stirring for 1 h at 0 °C and then the solvent volume was reduced. Cold $Et_2O$ was added, leading to the precipitation of the product which was filtered and washed with a few millilitres of cold EtOH. The resulting solid was dried *in a vacuum*.

Aspect: white solid.

Yield: 62 %.

m. p.: 200 °C (dec.).

El. Anal. Calcd. for $C_6H_{10}KNS_2$ (MW: 199.38 g·mol$^{-1}$): C, 36.14; H, 5.06; N, 7.03; S, 32.16%. Found: C, 36.10; H, 4.75; N, 6.88; S, 38.85%.

$^1$H-NMR ($D_2O$, 300.13 MHz, 298 K, $\delta$/ppm): 4.25 (tr, 4H, $H_{(1)} + H_{(5)}$); 1.60 (m, 6H, $H_{(2)} + H_{(3)} + H_{(4)}$).

### 4.2.2. Morpholine Dithiocarbamate Potassium Salt

Morpholine (23 mmol) and $CS_2$ (47 mmol) were added dropwise to an EtOH solution of KOH (23 mmol). The reaction mixture was stirred at 0 °C for 1 h. The solvent was removed and $CHCl_3$ was added to the product mixture, leading to the formation of a suspension of a white solid which was centrifugated and dried *in a vacuum*.

Aspect: white solid.

Yield: 81%.

m. p.: 300–320 °C (dec.).

El. Anal. Calcd. for $C_5H_8KNOS_2$ (MW: 201.34 g·mol$^{-1}$): C, 29.83; H, 4.00; N, 6.96; S, 31.85%. Found: C, 30.19; H, 4.01; N, 6.93: S, 33.59%.

$^1$H-NMR ($D_2O$, 300.13 MHz, 298 K, $\delta$/ppm): 4.35 (tr, br, 4H, $H_{(3)} + H_{(6)}$); 3.74 (tr, br, 4H, $H_{(2)} + H_{(5)}$).

### 4.2.3. Indoline Dithiocarbamate Sodium Salt (Na(indolineDTC))

NaH (26.81 mmol) was suspended in anhydrous THF under an inert atmosphere ($N_2$). Indoline (17.84 mmol) and $CS_2$ (excess) were sequentially added dropwise. The reaction mixture was stirred at 0 °C for 2.5 h. The suspension turned progressively from yellow to orange. Then, the mixture was centrifugated. The residual NaH deposited was quenched with MeOH, whereas the solution above was treated with hexane. A pale-yellow precipitate formed, which was isolated by centrifugation and dried in a vacuum.

Aspect: pale yellow solid.

Yield: 51%.

m. p.: 325 °C (dec.).

El. Anal. Calcd. for $C_9H_8NNaS_2$ (MW: 217.28 g·mol$^{-1}$): C, 49.75; H, 3.71; N, 6.45; S, 29.51%. Found: C, 47.82; H, 3.97; N, 6.16; S, 29.95%.

$^1$H-NMR ($D_2O$, 300.13 MHz, 298 K, $\delta$/ppm): 9.09 (m, 1H, aromatic H); 7.31 (m, 1H, aromatic H); 7.21 (m, 1H, aromatic H); 7.09 (m, 1H, aromatic H); 4.56 (tr, 1H, $H_{(2)}$); 3.06 (tr, 1H, $H_{(3)}$).

### 4.2.4. (3-(9H-Carbazol-9-yl)propyl)(methyl)dithiocarbamate Sodium Salt: Na(Carbz-pr-N(Me)-DTC)

Amine Precursor Synthesis: 3-(9H-Carbazol-9-yl)-*N*-methylpropan-1-amine(●HCl): Carbz-pr-NH(Me)(●HCl)

The synthesis of the amine precursor was conducted accordingly to a literature procedure [42]. Carbazole (Carbz; 10 mmol), tetrabutylammonium chloride (TBAB, 0.5 mmol) and 1,3-dibromopropane (30 mmol) were dissolved in benzene at 0 °C. Furthermore, 5 mL of aqueous NaOH 50% was added to the solution. The reaction mixture was stirred for 8 h at 0 °C and 12 h at room temperature. After that, the crude product was extracted with DCM (1 × 50 mL). The aqueous phase was washed with the same solvent (2 × 50 mL). The organic phases were joined together and the solvent was subsequently removed at a reduced pressure. A yellow oil was obtained. The oil was re-dissolved in MeCN jointly with $MeNH_2$●HCl (50 mmol) and $K_2CO_3$ (100 mmol). The mixture was stirred under reflux for 24 h and then the suspended solid was filtered off. The product was purified by silica

gel chromatography, using DCM/MeOH 9:1 as an eluent. Once the eluent had evaporated, the product was treated with a methanolic solution of HCl and Et$_2$O was added, leading to its precipitation.

Aspect: white solid.

Yield: 30%

Rf (on silica gel DCM/MeOH 9:1): 0.25.

Dithiocarbamate Synthesis: Na(Carbz-pr-N(Me)-DTC)

A total of 0.32 mmol of Cbz-pr-NH(Me)(•HCl) was dissolved in distilled water. The following reagents were subsequently added to the resulting solution: 0.64 mmol of aqueous NaOH and 0.64 mmol of CS$_2$. After the addition of NaOH, the precipitation of a white solid was observed. The reaction mixture was stirred at 0 °C for 7 h. Then, Ar was bubbled into the mixture for a few minutes and water was removed via lyophilization.

(*N.B.*: This Procedure Leads to a Product Comprehensive of 1 Equivalent of NaCl)

Aspect: white solid.

Yield: 96%.

El. Anal. Calcd for C$_{17}$H$_{17}$ClN$_2$Na$_2$S$_2$ (MW: 394.89 g·mol$^{-1}$): C, 51.71; H, 4.36; N, 7.09; S, 16.24%. Found: C, 45.81%; H, 4.36%; N, 6.02%, S, 13.43%.

$^1$H-NMR (D$_2$O, 300.13 MHz, 298 K, δ/ppm): 8.23 (m, 2H, H$_{(4)}$ + H$_{(5)}$); 7.64 (m, 4H, H$_{(2)}$ + H$_{(3)}$ + H$_{(6)}$ + H$_{(7)}$); 7.35 (m, 2H, H$_{(1)}$ + H$_{(8)}$); 4.47 (m, 2H, γ-CH$_2$); 4.14 (m, 2H, β-CH$_2$); 3.36 (s, 3H, CH$_3$); 2.26 (tr, 2H, α-CH$_2$).

Methyl(2-(naphtalen-2-ylamino)-2-oxoethyl)dithiocarbamate Potassium Salt: K(β-Napht-Sar-DTC)

Coupling (Z-SarOH)-(β-naphthylamine)

Under an inert atmosphere (N$_2$) at −15 °C (using a dry-ice bath in ethylene glycol), a solution of Z-SarOH (4.2 mmol) in anhydrous THF (solution A) and a solution containing β-naphthylamine (4.2 mmol), 4-methyl morpholine (NMM, 4.2 mmol) and isobutyl chloroformate (4.2 mmol) in the same solvent (solution B) were separately prepared. The solutions were stirred for 30 min and then solution A was transferred into solution B. After 30 min of continuous stirring with the same initial reaction conditions, the mixture was kept at 0 °C for 12 h (fridge). A white precipitate settled (NMMH$^+$), which was removed from the solution by centrifugation. Once the solvent was evaporated, the resulting solid was re-dissolved in ethyl-acetate. Two extractions were performed using 1 × HCl$_{(aq.)}$ 1M and 1 × NaHCO$_{3(aq.)}$ (saturated solution). Then, the organic phase was washed with brine and dried with Na$_2$SO$_4$, respectively. The isolated product was purified by silica gel chromatography, using DCM to elute byproducts and starting material's impurities and, afterwards, a mixture of DCM/MeOH 9:1 was added, which allowed for the collection of the desired species.

Aspect: white solid.

Yield: 35%.

$^1$H-NMR (CD$_2$Cl$_2$, 300.13 MHz, 298 K, δ/ppm): 8.17 (br, 1.5H, amide H); 7.89–7.21 (m, br, 12 H, aromatic Hs); 5.24 (s, 2H, N-CH$_2$-CO); 4.12 (s, 2H, Ph-CH$_2$-COO); 3.13 (s, 3H, CH$_3$).

Hydrogenation of the Coupling Product: Z-Protecting Group Cleavage

In a de-aerated round-bottom flask, the coupling product (1.46 mmol) was dissolved in DCM and a few mL of MeOH. Pd/C (10%; 51 mg) was added and H$_2$ (100%) was bubbled into the mixture for 18 h at room temperature under continuous stirring. The reaction progress was followed by TLC (DCM/MeOH 9:1 as eluent). The heterogeneous catalyst was filtered away with a celite filter and the solvent was removed was a pale-yellow oil, which was re-dissolved with MeOH, treated with a methanolic solution of HCl and, subsequently, with Et$_2$O. Precipitation of the chlorhydrate product took place, which was isolated by centrifugation.

Aspect: white solid.

Yield: 71%.

Rf (on silica gel DCM/MeOH 9:1): 0.35.

$^1$H-NMR (MeOH-$d_4$, 300.13 MHz, 298 K, $\delta$/ppm): 8.25 (s, 1H, aromatic H); 7.81 (q, 3 H, aromatic Hs); 7.60 (m, 1H, aromatic H); 7.44 (qui, 2H, aromatic Hs); 4.02 (s, 2H, CH$_2$); 2.81 (s, 3H, CH$_3$).

Dithiocarbamate Synthesis: K($\beta$-Napht-Sar-DTC)

The hydrogenated chlorhydrate product (0.879 mmol) was dissolved in H$_2$O. Then, 1.76 mmol of aqueous KOH was added leading to the formation of a suspended white solid. Next, 3.52 mmol of CS$_2$ was added. The reaction mixture was stirred overnight and lyophilized afterwards.

(*N.B.*: This Procedure Leads to a Product Comprehensive of 1 Equivalent of KCl)

Aspect: light brown solid.

Yield: 78%.

El. Anal. Calcd for C$_{17}$H$_{17}$ClN$_2$Na$_2$S$_2$(*KCl) (MW: 403.04 g·mol$^{-1}$): C, 41.42%; H, 3.25%; N,

6.96%; S, 15.91%. Found: C, 43.79%, H, 3.75%; N, 7.30%; S, 13.42%.

$^1$H-NMR (MeOH-$d_4$, 300.13 MHz, 298 K, $\delta$/ppm): 8.2–7.35 (m, 7H, aromatic Hs); 5.11 (s, 2H, CH$_2$); 3.66 (s, 3H, CH$_3$).

### 4.2.5. $\beta$-D-Glucoside-Conjugated Dithiocarbamate (gluc-MAE-DTC)

This ligand was synthesized and characterized as previously reported [43].

### *4.3. Synthesis of the Bimetallic Ruthenium Precursor*

The bimetallic ruthenium precursor was obtained as outlined in the results section and Figure 2.

### 4.3.1. [Ru(II)Cl$_2$(NBD)]$_n$

In a round-bottom flask, 8.21 mmol of RuCl$_3$(2H$_2$O) was dissolved in 100 mL of EtOH. Furthermore, 8.4 mL of NBD (821.1 mmol) was added under an inert atmosphere and the solution was refluxed (78 °C) for 3 days. A brown solid was isolated by centrifugation, filtered and washed with cold EtOH (2 mL, 3×) and benzene (2 mL, 3×). The product was dried under a vacuum overnight.

Yield: 70 %.

El. Anal. Calcd. (%) for C$_7$H$_8$Cl$_2$Ru ([Ru(II)Cl$_2$(NBD)], MW: 264.11 g·mol$^{-1}$): C, 31.84; H, 3.05; N, –; S –. Found: C, 32.50; H, 3.20; N, –; S –.

m. p.: 250 °C (dec.)

### 4.3.2. [Ru(II)(pipeDTC)$_2$(NBD)]

A total of 2.40 mmol of K(pipeDTC) was dissolved in 10 mL of DMF at 110 °C. A yellow solution formed, which turned light green within a few seconds. Then, 1.20 mmol of [Ru(II)Cl$_2$(NBD)]$_n$ was added to the solution, which was stirred for 20 min at room temperature. A yellow/brown suspension formed. The solid was centrifugated and washed with water (4x). The resulting yellow solid was dried in a vacuum (P$_2$O$_5$) and purified by column chromatography using DCM as eluent.

Aspect: yellow crystalline solid.

Yield: 39 %.

R. f. (on silica gel, DCM): 0.85.

El. Anal. Calcd. (%) for C$_{19}$H$_{28}$N$_2$RuS$_4$ ([Ru(pipeDTC)$_2$(NBD)], MW: 513.77 g·mol$^{-1}$): C, 44.42;

H, 5.49; N, 5.45; S 24.26. Found: C, 44.86; H, 5.15; N, 5.21; S 25.98.

ESI-MS *m/z* (100%), [Ru(pipeDTC)$_2$(DMDT)]$^+$ found (calculated): 514.02 (514.02)

### 4.4. Synthesis of Binuclear Heteroleptic Complexes of Ru(III)

All the complexes were obtained following the same route. The pipeDTC-oxidized precursor was dissolved in DCM at room temperature and *circa* 4 equivalents of a different dithiocarbamate pre-dissolved in a few millimetres of MeOH was dropwise added to the precursor solution. The solution was vigorously stirred for 10–15 min at RT. The solvent was removed and the product was re-dissolved in a minimal amount of DCM. The crude product mixture was filtered and was chromatographically purified using DCM/MeOH 95:5 in silica gel (chromatography conditions were only different in the case of the gluc-MAE-DTC heteroleptic derivatives). Once collected, the cleaned product was dried under vacuum in the presence of $P_2O_5$ and stored at 4 °C. The same procedure was also applied for the synthesis of the homoleptic $[Ru_2(pipeDTC)_5]Br$, which was used as a reference leading molecule in the evaluation of the cytotoxic activity of the complexes.

Since all the synthesized complexes were obtained from a non-pure bimetallic precursor, yields were not calculated. Similarly, only when a single heteroleptic derivative was obtained were the experimental elemental analysis reported among the theoretical values.

#### 4.4.1. $[Ru_2(pipeDTC)_5]Br$

Aspect: dark red solid.

m. p.: 226–235 °C (dec.).

R.f. (on silica gel, DCM/MeOH 95:5): 0.12.

El. Anal. Calcd. for $C_{30}H_{50}BrN_5Ru_2S_{10}$ ($[Ru_2(pipeDTC)_5]Br$, MW:1083.41 g·mol$^{-1}$): C, 33.26; H, 4.65; N, 6.46; S, 29.60%. Found: C, 32.35; H, 4.44; N, 6.15; S, 30.19%.

ESI-MS *m/z*, $[Ru_2(pipeDTC)_5]^+$ found (calculated): 1003.9365 (1003.9349).

#### 4.4.2. $[Ru_2(pipeDTC)_3(morphDTC)_2]Br$

Aspect: dark red solid.

m. p.: 237 °C (dec.).

R.f. (on silica gel, DCM/MeOH 95:5): 0.24.

El. Anal. Calcd. for $C_{28}H_{46}BrN_5O_2Ru_2S_{10}$ ($[Ru_2(pipeDTC)_3(morphDTC)_2]Br$; MW: 1087.35 g·mol$^{-1}$): C, 30.94; H, 4.26; N, 6.44; S, 29.48%. Found: C, 30.72; H, 4.18; N, 5.87; S, 30.48%.

ESI-MS *m/z*, $[Ru_2(pipeDTC)_3(morphDTC)_2]^+$ found (calculated): 1007.9023 (1007.8950).

#### 4.4.3. $[Ru_2(pipeDTC)_x(indolineDTC)_y]Br$

Aspect: dark red solid.

R.f. (on silica gel, DCM/MeOH 95:5): 0.20.

El. Anal. Found for $Ru_2(pipeDTC)_x(indolineDTC)_y]Br$: C, 35.75; H, 3.49; N, 5.52; S, 26.59%.

ESI-MS *m/z*, $[Ru_2(pipeDTC)_4(indolineDTC)]^+$, found (calculated): 1037.9205 (1037.9209).

ESI-MS *m/z*, $[Ru_2(pipeDTC)_3(indolineDCT)_2]^+$, found (calculated): 1071.9031 (1071.9054).

ESI-MS *m/z*, $[Ru_2(pipeDTC)_2(indolineDCT)_3]^+$, found (calculated): 1105.8823 (1105.8899).

#### 4.4.4. $[Ru_2(pipeDTC)_x(Carbz-pr-N(Me)-DTC)_y]Br$

Aspect: dark red solid.

R.f. (on silica gel, DCM/MeOH 95:5): 0.30.

El. Anal. Found for $Ru_2(pipeDTC)_x(Carbz-pr-N(Me)-DTC)_y]Br$: C, 41.21; H, 4.32; N, 6.36; S, 25.57%.

ESI-MS *m/z*, $[Ru_2(pipeDTC)_4(Carbz-pr-N(Me)-DTC)]^+$, found (calculated): 1156.9948 (1156.9969).

ESI-MS *m/z*, $[Ru_2(pipeDTC)_3(Carbz-pr-N(Me)-DTC)_2]^+$, found (calculated): 1310.0530 (1310.0536).

#### 4.4.5. $[Ru_2(pipeDTC)_x(β-Napht-Sar-DTC)_y]Br$

Aspect: dark red solid.

R.f. (on silica gel, DCM/MeOH 92:8): 0.55.

El. Anal. Found for Ru$_2$(pipeDTC)$_x$($\beta$-Napht-Sar-DTC)$_y$]Br: C, 39.28; H, 3.97; N, 6.68; S, 23.87%.

ESI-MS *m/z*, [Ru$_2$(pipeDTC)$_4$($\beta$-Napht-Sar-DTC)]$^+$, found (calculated): 1132.9583 (1132.9569).

ESI-MS *m/z*, [Ru$_2$(pipeDTC)$_3$($\beta$-Napht-Sar-DTC)$_2$]$^+$, found (calculated): 1261.9814 (1261.9784).

### 4.4.6. [Ru$_2$(pipeDTC)$_x$(DMDT)$_y$]Br

Aspect: dark red solid.

R.f. (on silica gel, DCM/MeOH 95:5): 0.15.

El. Anal. Found for Ru$_2$(pipeDTC)$_x$(DMDT)$_y$]Br: C, 28.76; H, 4.25; N, 6.40; S, 32.62%.

ESI-MS *m/z*, [Ru$_2$(pipeDTC)$_2$(DMDT)$_3$]$^+$, found (calculated): 883.8406 (883.8422).

ESI-MS *m/z*, [Ru$_2$(pipeDTC)$_3$(DMDT)$_2$]$^+$, found (calculated): 923.8732 (923.8736).

ESI-MS *m/z*, [Ru$_2$(pipeDTC)$_3$(DMDT)$_2$]$^+$, found (calculated): 963.9039 (963.9050).

### 4.4.7. [Ru$_2$(pipeDTC)$_x$(DEDT)$_y$]Br

Aspect: dark red solid.

R.f. (on silica gel, DCM/MeOH 95:5): 0.35.

El. Anal. Found for Ru$_2$(pipeDTC)$_x$(DEDT)$_y$]Br: C, 32.21; H, 4.49; N, 6.15; S, 30.34%.

ESI-MS *m/z*, [Ru$_2$(pipeDTC)$_3$(DEDT)$_2$]$^+$, found (calculated): 979.9362 (979.9364).

ESI-MS *m/z*, [Ru$_2$(pipeDTC)$_4$(DEDT)]$^+$, found (calculated): 991.9379 (991.9365).

[Ru$_2$(pipeDTC)$_3$(gluc-MAE-DTC)$_2$]Br and [Ru$_2$(pipeDTC)$_4$(gluc-MAE-DTC)]Br

[Ru$_2$(pipeDTC)$_x$(gluc-MAE-DTC)$_y$]Br

Two reaction products were separately collected by inverse chromatography.:

[Ru$_2$(pipeDTC)$_4$(gluc-MAE-DTC)]Br and [Ru$_2$(pipeDTC)$_3$(gluc-MAE-DTC)$_2$]Br.

Purification: C$_{18}$ column; strong eluent: MeCN/H$_2$O 9:1; weak eluent: H$_2$O. The strong eluent percentage was raised from 5% to 15% in 20 min, then from 15% to 50% in 30 min. R.f. ([Ru$_2$(pipeDTC)$_3$(gluc-MAE-DTC)$_2$]Br; on silica gel, DCM/MeOH 9:1): 0.25;

R.f. ([Ru$_2$(pipeDTC)$_4$(gluc-MAE-DTC)]Br; on silica gel, DCM/MeOH 9:1): 0.20.

### 4.4.8. [Ru$_2$(pipeDTC)$_3$(gluc-MAE-DTC)$_2$]Br

Aspect [Ru$_2$(pipeDTC)$_3$(gluc-MAE-DTC)$_2$]Br: red solid.

m. p.([Ru$_2$(pipeDTC)$_3$(gluc-MAE-DTC)$_2$]Br)): ~143 °C.

El. Anal. Calcd. for C$_{38}$H$_{66}$BrN$_5$O$_{12}$Ru$_2$S$_{10}$ ([Ru$_2$(pipeDTC)$_3$(gluc-MAE-DTC)$_2$]Br, MW: 1387.63 g·mol$^{-1}$): C, 32.89; H, 4.79; N, 5.05; S, 23.10%. Found: C, 32.38; H, 4.35; N, 4.44; S, 22.74%.

ESI-MS *m/z*, [Ru$_2$(pipeDTC)$_3$(gluc-MAE-DTC)$_2$]$^+$ found (calculated): 1308.0011 (1307.9977).

### 4.5. Viability Studies

The MDA-MB-231 or PC3 cells were seeded in 96-well plates and treated after 24 h with [Ru$_2$(pipeDTC)$_5$]Br, [Ru$_2$(pipeDTC)$_3$(morphDTC)$_2$]Br, or [Ru$_2$(pipeDTC)$_3$(gluc-MAE-DTC)$_2$]Br to the final concentration of 0.05 µM, 0.1 µM, 1.5 µM, 5 µM, and 10 µM, respectively. The complexes were dissolved in DMSO. The IC$_{50}$ values were calculated 24 and 48 h after treatment from cellular viability traces obtained using a Trypan blue test or WST-1 test. Results were normalized to the control (vehicle). Three independent experiments were carried out for each analysis in triplicate. The full experimental procedure is presented in SI.

### 4.6. Cell Cycle Studies

The MDA-MB-231 or PC3 were treated with [Ru$_2$(pipeDTC)$_5$]Br, [Ru$_2$(pipeDTC)$_3$(morphDTC)$_2$]Br, or [Ru$_2$(pipeDTC)$_3$(gluc-MAE-DTC)$_2$]Br to the final concentrations of 0.9 µM, 1.3 µM, and 20 µM, respectively. Immediately after treatment, cells were detached and incubated on ice with propidium iodide for 20 min. The same procedure was followed

to study the effect of the complexes on the cellular cycle 24 and 48 h after treatment, respectively. Following incubation with propidium iodide, cells were further analysed using a cytofluorimeter. At least $10 \cdot 10^3$ events were studied for each sample. The full experimental procedure is available in SI.

## 5. Conclusions

To the best of our knowledge, no previous attempts to isolate heteroleptic binuclear ruthenium dithiocarbamate complexes have been made. Provided that our synthetic strategy for the synthesis of pure bimetallic dithiocarbamate Ru(III) species still requires optimization, we believe this work will serve as a good basis for further advancements. Moreover, the preliminary antiproliferative studies of compounds #2 and #3 are encouraging and serve as a good proof of concept that bimetallic ruthenium dithiocarbamate complexes may retain the cytotoxicity of homoleptic analogues, but can also introduce additional features that make them promising candidates in the development of tuneable anti-cancer agents.

**Supplementary Materials:** The following supporting information can be downloaded at: https://www.mdpi.com/article/10.3390/inorganics10030037/s1.

**Author Contributions:** Conceptualization, A.E.G., L.B. and D.F.; methodology, A.E.G., L.B. and N.P.; investigation, A.E.G. and L.B., formal analysis, A.E.G., L.B., N.P. and D.F.; data curation, A.E.G.; writing—original draft preparation, A.E.G. and D.F.; writing—review and editing, A.E.G. and D.F.; supervision, D.F. All authors have read and agreed to the published version of the manuscript.

**Funding:** This research was funded by A.R.TE.M.O. ONLUS Association (via Campolongo 4E Padova, Italy, N° Registro 3524, C.F. 93046520255. A PhD fellowship was granted to N.P. by T.R.N. IMBALLAGGI—logistics services, 29 February 2016, Prot. 539, Tit. III, Cl.13, Fasc. 3. A.E.G. is grateful to the Engineering and Physical Sciences Research Council for a PhD studentship. The funders had no role in the design of the study; in the collection, analyses, or interpretation of data; in the writing of the manuscript, or in the decision to publish the results.

**Data Availability Statement:** All data is available in this manuscript or the supporting information.

**Acknowledgments:** The authors are grateful to DIVAL Toscana for conducting the cell viability and cell cycle experiments.

**Conflicts of Interest:** The authors declare no conflict of interest.

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
