# Peer review of "Binuclear Heteroleptic Ru(III) Dithiocarbamate Complexes: A Step towards Tunable Antiproliferative Agents"

_inorganics, doi:10.3390/inorganics10030037_

Round 1

Reviewer 1 Report

The paper of Dolores Fregona and co-authors is fundamental work on synthesis new Ru(III) heteroleptic dithiocarbamate complexes. In present work was proposed a convenient method for the synthesis of heteroleptic complexes, their biological activity was studied.

I have questions:

  1. Despite the abundance of synthesized compounds, none of them has been isolated in crystalline form. When dealing with heteroleptic complexes, it is very important to grow a crystal, establish its structure, and then do an X-ray powder diffraction pattern to prove that you do not have homoleptic complex impurities. Moreover, from the earlier works by the same authors (Chemistry 2012, 18, 14464-14472 or Dalton Trans 2018, 47, 15477-15486) we know that the authors are able to grow crystals and they have the ability to determine the structure. So, what was the problem to obtain crystals? With the X-Ray this paper would be brilliant!

  1. Due to the lack of a structure and a powder diffraction pattern, questions arise about the purity of the compounds obtained. And towards the end of the article, the authors themselves confirm this: (lines 438-439) “…only the morphDTC and gluc-MAE-DTC ligands allowed for the obtainment of stoichiometrically pure products…”. If pure products are isolated for only four compounds out of the seven described in the experimental part, then - why was the IR spectrum and NMR spectrum taken from samples with impurities?

  1. Since the authors rely too much on electrospray mass spectrometry, I suggest preparing a sample containing a stoichiometric mixture of homoleptic complexes. And in the spectrum it will be possible to notice the formation of a heteroleptic complex. I believe that the authors should have done a similar experiment while doing the work, it would be very interesting to see the result.

  1. Line 40-41 I suppose that authors are talking about ruthenium dithiocarbamate complexes because the CCDC contains examples of ruthenium heteroleptic compounds.

  1. Line 179-182. This is a very interesting remark. Do the authors think that the IR spectrum of a heteroleptic complex and a stoichiometric mixture of homoleptic complexes will be very different? Having such evidence would greatly improve the part about IR spectroscopy.

  1. About H-NMR, for example, Figure 8 and lines 275-279. The same suggestion - to make an NMR spectrum of a stoichiometric mixture of [Ru2(DMDT)5]Br and [Ru2(pipeDTC)5]Br in order to see that the missing peaks (discussed in the paper) in the resulting spectrum will be present. And the resulting spectrum will differ from the spectrum obtained for the heteroleptic complex [Ru2(pipeDTC)x(DMDT)y]Br.

  1. In biological part, there is also a lack of the biological activity of a stoichiometric mixture of homoleptic complexes. Suddenly, a simple mixture of homoleptic complexes will have the same activity as heteroleptic ones, then it will be enough just to mix two complexes with a well-established synthesis technique instead of synthesizing heteroleptic complexes.

And recommendation

  1. Since the authors isolated only three products in their pure form, I advise to remove the synthesis description of heteroleptic complexes with unconfirmed purity in supplementary materials.

Despite my comments and doubts about the existence of the obtained heteroleptic complexes, I advise to accept this article in Inorganics after answering my questions, so that other researchers can have the opportunity to discuss this topic.

Reviewer 2 Report

This paper describes a modified protocol for the synthesis of bimetallic dithiocarbamate Ru(III) complexes and investigation of their antiproliferative activity. The complexes presented were characterized by different methods (ESI-MS, NMR, FT-IR, and UV-Vis) to confirm their formulations and evaluate coordination patterns of the metal and ligands. In general, the work seems to be well performed. Thus, I recommend the acceptance of this manuscript after addressing the issues listed below. 
1. The X-ray structures of the compounds obtained are absent. Is it possible to grow single crystals in order to perform X-ray structure determination? Structural data would greatly enhance the significance of this work. 
2. The abstract contains only one sentence concerning the essence of this work. It would probably be better to make the abstract more reflective of the essence of the work done.

Reviewer 3 Report

Comments are listed in the attached word file.

Round 2

Reviewer 1 Report

Thanks dear authors for answers to questions. 

We are engaged in the synthesis of heteroleptic complexes of lanthanides and Pt/Pd with various ligands. And the experiment with a stoichiometric mixture proposed by me in each method of analysis makes sense. In MALDI, homoligand complexes can be mixed and ions from the heteroleptic complex can be seen.

And a stoichiometric mixture of platinum homoligand complexes shows the same activity against cancer cells as a heteroleptic complex (in our case. I don't know anything about yours). Therefore, I advised you to give the next time for analysis also a stoichiometric mixture of homoligand complexes. An additional experiment will decorate your work.

Good luck!

Reviewer 2 Report

The authors have addresed my comments. Thus, the manuscript can now be accepted.